# Benefit of an Ultrasonic Irradiation on the Depollution by Washing of Nickel- or Zinc-Contaminated Vermiculite

**DOI:** 10.3390/molecules30051110

**Published:** 2025-02-28

**Authors:** Antoine Leybros, Sophie Herr, Rita Salameh, Rachel Pflieger

**Affiliations:** 1CEA, DES, ISEC, DMRC, Univ Montpellier, 30207 Marcoule, Francerita.salameh@cea.fr (R.S.); 2ICSM, Univ Montpellier, CEA, CNRS, ENSCM, 30207 Marcoule, France

**Keywords:** soil washing, vermiculite, sequential extraction, ultrasound, nickel, zinc, heavy metal

## Abstract

Pollution of soil by heavy metals has become a critical environmental issue. This study investigated an innovative approach to heavy metals removal, focusing on the desorption of nickel and zinc from vermiculite using a combination of leaching and ultrasonic (US) irradiation at 20 or 362 kHz. When 0.1 M HCl was used as a washing solution, Zn^2+^ desorption yields around 85% were obtained in all conditions. Under 20 kHz US, fragmentation of the particles occurred, leading to the formation of new sites where released Zn^2+^ could sorb, allowing improved decontamination by cation exchange. Even higher yields were obtained with the biobased citric acid. Ni^2+^ desorption yields were lower due to its distribution in less accessible Tessier fractions. They significantly increased under US, especially at 362 kHz. It is shown that US leads to transfer of the contaminant from less accessible fractions (in particular the residual one) to more accessible ones, and that at low frequency, new sorption sites are created by fragmentation, leading to readsorption in the exchangeable fraction. This study brought to light for the first time the potential of high-frequency US in enhancing soil washing, to a higher extent compared to low-frequency (20–50 kHz) US.

## 1. Introduction

As reflected in many of the goals of the 2030 Agenda for Sustainable Development at the United Nations Summit [1], soil and groundwater pollution has increasingly become a global environmental issue, due to the development of anthropogenic and industrial activity. Over 5 million sites with soil polluted with toxic elements have been inventoried throughout the world [2]. Amongst these elements, heavy metals represent around 35% of the identified pollutants affecting soil in Europe [3]. If heavy metals are naturally present in soils at trace levels due to geological weathering, accumulation occurs above threshold values causing risks to human, biomass and ecosystems in rural and urban environments [4]. Amongst these heavy metals, nickel and zinc contamination can originate from a wide variety of industrial or agricultural activities such as land application of fertilizers, animal manures and pesticides, mining, waste disposal, etc. [5,6].

In this context, it is necessary to handle site depollution with a view to minimizing hazardous effects and rehabilitating soils. Many soil remediation processes have already been applied, divided into stabilisation and extraction methods, depending on the objective: to fix contamination, reduce metal migration capacity or extract it [7,8,9]. Despite their efficiency to contain metal pollution and limit their dispersion, stabilization techniques do not allow to obtain cleaned soils, able to fulfil their primary function [10]. Thermal treatments have high energy and financial costs and provoke disruption of biological and physicochemical soil properties [11]. As for bio- and phyto-remediation techniques, they exhibit lower treatment kinetics [12,13]. Consequently, leaching with relevant chemical agents [14] is emerging as a promising solution to extract heavy metals with high extraction yields, short extraction kinetics and low operating costs, while respecting biological and physicochemical characteristics of soils [15].

Most soil leaching processes described in the literature require a low pH to maintain heavy metal pollutants in ionic form and make metal desorption easier since hydrogen ions compete with them to adsorb on sites [16]. In this way, HCl was often considered as a cheap and efficient leachant. Different phenomena may occur depending on the acid concentration, such as ion exchange, precipitation or dissolution.

As the metals may be present in different forms, their speciation or at least distribution in the different fractions of the solid matrix must be considered to determine metal mobility and assess technical feasibility. To obtain some better insight into the element association with minerals or organic matter, its mobility and plant availability, several sequential extraction schemes were proposed, the most widely used ones being that of Tessier et al. (Tessier 1979) [17] and that developed by the European Commission Standards, Measurements and Testing Programme (BCR) (BCR 1996). [18] In the successive steps, the applied chemical treatment goes from mild to harsh, quantifying separately contaminants bound weakly to strongly to the solid phase. In this respect, sequential chemical extraction is an operational speciation [19,20]; the species is defined by the chemical conditions in which it is extracted. Obviously, the weakly bound elements will have higher potential mobility into the environment. In their 1979 work, Tessier et al. proposed a sequence of chemical extractions to quantify separately (successively) the metal present in the most common constituents of sediments, each constituent being affected differently by environmental conditions. The first step quantifies the exchangeable cations (F1), corresponding to cations adsorbed on clays, hydrated iron or manganese oxides and humic acids. The second fraction (F2) refers to metals easily released at acidic pH (acetic acid extractable fraction), the third one under anoxic (reducing) conditions and the fourth one under oxidizing conditions. The residual fraction F5 is determined by comparison with the total metal concentration. Tessier’s sequential extraction procedure remains one of the mostly used sequential extraction procedures, for sediments but also for soils.

Sequential extraction tests were carried out to determine the fraction of soil from which the metal was removed, for leaching [21,22,23], and also for other remediation techniques [24,25]. Moutsatsou et al. [26] screened inorganic acids for the treatment of soils polluted due to mining activities. The best desorption yields were obtained using HCl with a concentration of 2 M and a soil/acid ratio equal to 1:20 g·mL^−1^. HNO_3_ hindered zinc mobility due to its oxidizing nature and the formation of insoluble zinc nitrate salts. Zinc extraction from soils was demonstrated as possible but improvable with yields ranging between 67 and 92% according to operating conditions and site characteristics [27,28]. Organic acids, such as, in a non-exhaustive way, oxalic acid, tartaric acid, citric acid or ethylenediaminetetraacetic acid, have also been used as an alternative to inorganic acids for heavy metals complexation while maintaining a low pH of the washing solution. Using these biocompatible organic acids, alone or in combination, can allow to reach similar heavy metal removal yields as inorganic acids, depending on soil nature and operating conditions [26,29,30,31]. Citric acid is one of the compounds with the highest ability to complex Zn and Ni [32]. A zinc removal yield around 62% was obtained after 30 min treatment on a polluted agricultural soil with a citric acid concentration of 0.1 M and a soil/washing solution ratio equal to 1:20 g·L^−1^ [33].

A combination of leaching with other techniques may contribute to improve heavy metal desorption from soils. Among them, leaching assisted with ultrasound was shown to increase desorption yields for a given treatment time in many cases (but not for all elements nor for all washing solutions [34]). For instance, Kim et al. [33] and Choi [35] showed that the application of low-frequency ultrasound with mechanical stirring in the HCl washing of soil allowed to reach zinc extraction yields of 38–47.9%, compared to 10–18.1% with mechanical stirring only. Similarly, Park [36] coupled sonication (at 28 kHz) to mechanical stirring, and reported increased desorption yields compared to mechanical stirring alone: 76.2% vs. 42.2% for Cu, 75.4% vs. 46.9% for Pb and 72.0% vs. 46.3% for Zn, for a 30 min treatment in 1 M HCl. Similar beneficial effects of ultrasound were observed in the desorption of Hg^2+^ under acidic conditions: 33% with 28 kHz US irradiation, vs. 8% without [37].

These higher yields at given times allow shorter treatment duration and smaller secondary wastewater generation. The synergy of leaching, mechanical stirring and ultrasonic irradiation was underlined [33] and explained by the formation of cavitation bubbles under US, whose collapse generates a strong mixing at the microscale via the emission of microjets and shockwaves. It is yet to be underlined that sonication alone leads to low desorption yields, due to very limited ultrasound (US) propagation in viscous media. Interestingly, in recent works [38,39], sonication was also reported to modify the distribution of the metal contaminants in the soil fractions, and in particular to trigger the transfer of Pb and Zn from oxidisable and residual fractions to acid extractable and reducible fractions.

The aim of the present study is to further evaluate how the contribution of US may be beneficial for soil remediation by leaching. All studies that combined US irradiation with soil washing worked with low-frequency US, either with a 20 kHz sonotrode or with a 40–50 kHz ultrasonic bath, probably because these conditions enhance the physical effects of US and are easily scalable [40]. However, it appears interesting to look also at higher frequencies that would cause less damage to the soil structure. A second lack in most existing studies is the absence of monitoring of the depollution kinetics. Without it, it cannot be concluded whether US increases the attainable desorption yield or only accelerates the depollution.

Considering the strong disparity of soil composition according to sampling, it is necessary to choose a model matrix. Vermiculite was chosen, as a phyllosilicate widespread in rocks, sediments and soils [41,42], with high capacity towards nickel and zinc [43,44], considered here as representative heavy metal pollutants. The desorption of nickel and zinc, two heavy metals with significantly different behaviours in terms of interaction with soils, was studied implementing a strong acid commonly used in the literature, HCl, and an organic acid with complexation properties, citric acid. HCl concentration was set to 0.1 M, as a compromise between efficiency and minimization of the impacts on the soil nature. [45]

Washing was performed in silent mode (i.e., under stirring, without US) and with US irradiation at two different frequencies: 20 and 362 kHz. The high frequency was chosen based on the laboratory previous experience: high enough not to damage the soil structure but low enough to limit wave absorption by the viscous medium (the wavelength is inversely proportional to the frequency); it is also around the optimum for chemical activity [46]. The coupling of acid leaching with Mg^2+^ ion exchange was also considered to prevent Ni or Zn readsorption. The distribution of Ni and Zn in the different fractions of vermiculite was studied using the Tessier sequential protocol before and after treatment, to assess heavy metal mobility.

## 2. Results and Discussion

### 2.1. Characterization of Metal-Loaded Vermiculite

Figure 1 presents the kinetics and isotherms of Zn and Ni sorption on vermiculite. Looking at the kinetics curves, equilibrium is reached within 3 days and the sorption capacity in the tested conditions (C_0_ = 100 mg/L) is around 35–40 mg·g^−1^ for both metal contaminants. Based on these results, adsorption isotherms were established using the protocol described in Section 3.3. With increasing Zn^2+^ or Ni^2+^ concentration in solution, sorption sites gradually become saturated. A similar behaviour is noticed for both metals with maximum adsorption capacities around 95 mg·g^−1^ for Zn and 55 mg·g^−1^ for Ni. The pH varies from 4 to 6–7 as sorption proceeds.

The initial distribution of the contaminant in vermiculite was quantified for each batch using the Tessier sequential extraction protocol. Their average is presented in Figure 2. A marked difference is observed between both metals. While Zn is found mainly (57%) in the F1 and F2 fractions, indicating that it is easily removable in acidic media, these two fractions account for only 20% of the sorbed Ni, while approx. 38% of Ni is in the residual fraction, followed by the F3 (25%) and F4 (17%) fractions. The removal of Ni is thus expected to require harsher conditions. This difference in the behaviour of Ni and Zn is in agreement with the literature [47,48,49]. Some variability was observed between the batches, translating into large standard deviations in Figure 2. To account for it, all subsequent comparisons with the initial state were performed within the same batch.

Here it can be noted that the F2 fraction is often called the “carbonate” fraction, following Tessier’s nomenclature and because many samples are carbonate-rich. This is, however, not the case here, with a carbon amount estimated to 0.45 mg/g by carbon analyser—this low content cannot account for both the F2 and F4 (“organic”) fractions. This improper naming is well known and it is in general recommended to call Tessier extraction protocol fractions by their chemical properties rather than probable minerals [19,20]. In the particular case of the F2 fraction, it was for instance shown by Parat et al. on an acidic sandy soil [50] that sodium acetate did not only remove exchangeable and carbonate-bound Zn, but also very large amounts of this metal in other forms. This non-selectivity was confirmed by Hanahan [51] who showed that sodium acetate could also release copper and lead associated with hydroxide minerals.

Contrary to F2, however, the presence of iron in the chemical formula of the vermiculite may indicate that Zn and Ni found in the F3 fraction correspond to metal bound to iron oxides.

Figure 3 compares XRD patterns of initial, Zn-contaminated and Ni-contaminated vermiculite samples.

The XRD pattern of original vermiculite exhibits major characteristic peaks in the low-angle region at 2θ = 5.9°, 6.9°, 7.3° and 8.7° corresponding to different interlayer structures with respective interlayer spacings of 14.7, 12.6, 12.1 and 10.1 Å. Upon contamination with Ni^2+^ or Zn^2+^ cations, the peak at 29.4° disappears, and peaks appear at 2θ = 12°, 25° and 45°. The shift of the peak at 5.9° for raw vermiculite to 6.1° for contaminated samples may be attributed to interlayer cation exchange with K^+^ or Ca^2+^ cations. The slight changes of XRD patterns in the 5–10° 2θ region may indicate that only small amounts of Ni^2+^ or Zn^2+^ are sorbed in the interlayer spacing and that the majority of them may be on the surface or at the edges of vermiculite.

### 2.2. Zn/Ni Desorption by Washing with 0.1 M Hydrochloric Acid Solution

#### 2.2.1. Desorption Kinetics

Appendix A summarizes all obtained results, allowing to highlight the influence of operating parameters such as US frequency (silent conditions, 20 and 362 kHz) and ratio of vermiculite mass to HCl solution volume (m/V ratio) on Ni and Zn extraction kinetics and yields.

Looking at Zn desorption at an m/V ratio equal to 20 g·L^−1^, Figure 4a shows that there is no big influence of US irradiation, whatever the US frequency, on extraction yields or kinetics. It just seems that irradiation with 362 kHz US may lead to a slightly higher desorption yield. Whatever the conditions, desorption is very fast (a plateau is reached within 15 min) and extraction yields are around 75% after 1 h duration. Results obtained for Ni^2+^ desorption in the same conditions are significantly different (Figure 4b). Extraction kinetics are much slower and no plateau appears, even after 3 h duration. Ni extraction yields are significantly lower than Zn ones, reaching 14–21% after 1 h. These results are in agreement with those of Kuo et al. [45] on metal-contaminated rice soils. They showed that a higher HCl concentration was needed to desorb a higher quantity of Ni. Interestingly, in Figure 4 it appears that the impact of adding an ultrasonic irradiation depends on the US frequency. While at 362 kHz the extraction yield increases (29% after 3 h, compared to 24% in V-silent conditions), it decreases when 20 kHz US is applied or in the horizontal shaking mode. Such results can be explained by the evolution of the particle size distribution and of Ni repartition in Tessier fractions, as discussed below.

Another key process parameter to consider in a view to process sizing and scale-up is the m/V ratio. This ratio has been varied from 10 to 100 g/L. As exemplified in Figure 5a–f, the higher the ratio, the lower the extraction, in agreement with the literature [38]. For Zn, this decrease is observed starting from 50 g/L, for Ni already at 20 g/L. This decrease may be due to a lower relative number of protons at higher vermiculite mass (higher desorption yields are obtained at higher acid concentration [52]), and to a higher viscosity that limits collisions between particles and diffusion. The decrease in the extraction yield is more pronounced under high-frequency US, because the presence of particles and a higher viscosity restrain the propagation of the US wave [53], so that only a smaller zone of the suspension will be effectively submitted to cavitation bubbles. This effect explains why at 20 g/L the highest yields are obtained with 362 kHz irradiation, while at 50 g/L higher yields are obtained with 20 kHz and horizontal stirring. The effect is less pronounced at 20 kHz because of the larger wavelength (the wavelength is inversely proportional to the frequency) and of the strong mixing induced by low-frequency US [40].

#### 2.2.2. Particle Size Distributions

Particle size distributions of vermiculite after Ni or Zn contamination (there is no impact of chemical pollutant nature on this distribution) and after silent and ultrasonic desorption are given in Figure 6 for various m/V ratios. Only very small changes are observed under V-silent conditions and under 362 kHz US irradiation. Only a slight decrease in larger particles (>100 μm) fraction is noticeable, due to disagglomeration. Under H-silent conditions, a tendency to decrease in size is observed, particularly visible in the range 10–50 µm, and more pronounced for higher m/V ratios. This decrease can be attributed to fragmentation induced by collisions between particles, collisions whose probability increases with the m/V ratio. The size distributions obtained after 20 kHz irradiation are shifted towards smaller sizes: particles are fragmented due to the mechanical effects of low-frequency ultrasound (microjets and shockwaves) [54,55]. Here the impact of the m/V ratio is limited: while a decrease in size is observed from 10 to 20 g/L, the distribution is on the contrary shifted towards bigger sizes for 50 g/L. This may be accounted for by reduced cavitation effects at higher viscosity [53].

#### 2.2.3. Tessier Sequential Extractions

The Tessier sequential extraction protocol was carried out on vermiculite samples after washing in 0.1 mol/L HCl under silent conditions and under US irradiation. Figure 7 exemplifies the metal repartition in the different fractions for (a,b) Zn and (c) Ni, for an m/V ratio of 20 or 50 g/L. In each case, the dashed grey fraction corresponds to the metal removed by the treatment. Results obtained under V-silent conditions are not plotted but are very close to those at 362 kHz. In all conditions, fractions F2–F4, which initially amount to 57% (resp. 77%) of the present Zn in Batch 3.5G(Zn) (resp. 3.2), are reduced to almost zero after washing, which explains the very high Zn desorption yields obtained (Figure 4a). The residual fraction (F5) increases in H-silent conditions, which may indicate readsorption of the Zn in solution on strongly binding sites. It stays constant at 20 kHz and decreases at 362 kHz and under V-silent conditions. This confirms that sonication can trigger the transfer of metal pollutants from the residual fraction to more accessible ones [38,39]. Comparing the two US frequencies, it seems, although not intuitive, that fragmentation (higher at 20 kHz and H-silent) is not favourable to the desorption of strongly bound or not easily accessible Zn. On the other hand, the F1 fraction (exchangeable cations) decreases in all cases.

Increasing the m/V ratio to 50 g/L enhances the differences observed between the experimental conditions. The decrease in the F5 fraction under V-silent and 362 kHz conditions here leads to an almost zero fraction, while this fraction slightly increases under 20 kHz and H-silent, i.e., under conditions that favour fragmentation of the vermiculite particles. Possibly, fragmentation leads to the creation of new sites of strong Zn bonding. As for the F1 fraction, at 50 g/L it increases in all conditions. This increase may be explained by newly created surfaces presenting sorption sites for exchangeable cations and above all by F1 sites sorbing cations released from F5 (hence, the higher increase for 362 kHz and V-silent conditions).

The Ni case shows very different trends. First of all, a non-negligible amount of nickel is still measured in the F2, F3 and F4 fractions after HCl washing, which might look surprising, because these fractions are sensitive to acidic conditions, in particular the F2 fraction. This presence indicates probable redistribution of the metal contaminant in the different fractions during the sequential extraction protocol. Second, while the F1 fraction was shown to decrease in all cases for Zn, for Ni it stays constant (around 14%) for 362 kHz irradiation and H-silent conditions and increases from 11% to 17–19% at 20 kHz. As for the F5 fraction, under H-silent conditions it increases, similarly to the Zn case, from 29 to 42%, probably due to readsorption of the solubilized Ni on strongly binding sites. It stays constant (29–31%) at 362 kHz and decreases from 46 to 30–31% at 20 kHz. Sonication thus either limits readsorption onto the F5 fraction, or allows to desorb part of the Ni bound in it, thanks to microjets and shock-waves increasing the solution penetration into the pores or to fragmentation that renders new sites accessible. The redistribution of Ni in the different fractions is particularly marked at 20 kHz, frequency that leads to intense fragmentation and consequently to creation of new adsorption sites. Interestingly, increasing the m/V ratio from 20 to 50 g/L does not lead to any change in the Ni distribution. The large remaining F5 fraction combined with the Ni redistributed in the F2-F4 fractions is in agreement with the low desorption yields measured.

### 2.3. Improving Zn/Ni Desorption by Coupling HCl Acidic Washing with Ion Exchange

In general, applying 20 or 362 kHz frequency ultrasound has allowed to reduce the F5 residual fraction. For Ni at 20 kHz this decrease is accompanied by an increase in the F1 exchangeable fraction. In all cases, a significant proportion of Zn or Ni is still present on exchangeable sites after treatment, probably due to sorption of metal released from the F2, F3 or F4 fractions. Chemical bonds involved in the F1 fraction are weak electrostatic interactions. Therefore, an ion exchange with divalent cations may increase extraction, as indicated in existing sequential protocol [56]. A simultaneous combination of 0.1 M HCl acidic washing with ion exchange using 1 M MgCl_2_ solution has thus been applied. At such a high concentration, it is expected that Mg^2+^ ions will help desorb Zn^2+^/Ni^2+^ cations initially present in the F1 fraction, and sorb preferentially onto the newly created sites.

The desorption kinetics observed with the mixed washing solution (not shown here) is very fast, similarly to that with HCl. Figure 8 shows Zn distribution in vermiculite depolluted by this method in V-silent conditions and with the application of 362 kHz US, at an m/V ratio of 20 g·L^−1^. As seen on the example of the 362 kHz treatment, the coupling of acidic washing and ionic exchange allows to increase Zn desorption yield from 78% to 86% compared to HCl washing alone. This increase is due to the expected drastic reduction in the F1 exchangeable fraction (to 2%, compared to 12% with HCl alone). The same final yield and decrease in the F1 fraction are observed under V-silent conditions.

The same experimental protocol has been carried out on Ni-polluted vermiculite. First, Ni desorption kinetics have been monitored at an m/V of 20 g·L^−1^ in V-silent conditions, with 20 kHz and with 362 kHz US irradiation (Figure 9). In V-silent conditions, Ni desorption seems to be slightly less efficient under the coupling of acid washing and ion exchange than with the sole acid washing (20% vs. 24% extraction yield)—though the difference is within the experimental uncertainty. The kinetic behaviour is the same in both cases. Under 362 kHz US irradiation the desorption under MgCl_2_-HCl washing is faster in the first half an hour, then the two curves are similar. On the contrary, at 20 kHz there is a significant gain when coupling acid washing and ion exchange: Ni desorption yield increases from 21 to 38%. This promising result may be attributed to particle fragmentation caused by low-frequency ultrasound, which induces an easier Ni release and the formation of more cation exchange sites.

Corresponding Ni Tessier fraction diagrams are plotted in Figure 10. When applying US at 362 kHz frequency (Figure 10a), for both washing solutions the F2 fraction almost disappears, the F3 fraction sharply decreases (from 26 to 9% with HCl, 28 to 10% with HCl + MgCl_2_) and no significant evolution is noticeable for the F4 and F5 fractions. The exchangeable F1 fraction has been reduced more in the presence of MgCl_2_ (from 8 to 4%, vs. from 14 to 12% with HCl alone). Probably, the release of exchangeable cations occurs faster in the presence of MgCl_2_, which would explain the faster desorption observed in the first half an hour. The still similar desorption yields obtained at 362 kHz with or without MgCl_2_ can be accounted for by the small values of F1 exchangeable fractions, which leave too little margin to improve the global process. Finally, the best results obtained at 20 kHz US frequency are explained by considering the Tessier distribution diagram in Figure 10b. The large fragmentation of particles induced by this low frequency has led to the formation of new adsorption sites, leading, under HCl washing, to an increase in the F1 fraction. The presence of Mg^2+^ cation has prevented the readsorption of Ni^2+^ on the created exchangeable sites, leading to a strong reduction in F1 (to 4%). This decrease combined with the significant (by a third) reduction in the F5 residual fraction, contrary to other cases, explains the better yield observed.

To further improve Ni removal from polluted vermiculite, other leachants or complexing agents may be considered in coupling with sonication. Operating conditions (vermiculite mass; solution concentration and volume) have to be optimized to promote mobilization of Ni in the F1 exchangeable fraction. MgCl_2_ concentration will then be defined accordingly in terms of management of secondary effluents to be treated.

### 2.4. Metal Desorption Using Citric Acid

Citric acid is a triacid with acidity constants of 3.13, 4.76 and 6.40. It can form negative complexes with heavy metal ions such as Zn^2+^ and Ni^2+^ [57]. Washing experiments were performed at 10 g/L vermiculite, contaminated at 38 mg/g Zn^2+^ (resp. 35 mg/g Ni^2+^), which corresponds to a maximum metal concentration in solution of 6 mM. The citric acid concentration was chosen at 0.5 M to ensure large excess and complexation of all metal present in solution and the pH was fixed at environmentally friendly values: between pH 5.5 and 7. Appendix A shows speciation diagrams of Zn^2+^ and Ni^2+^ calculated using PhreeqC for a solution containing 380 mg/L Zn^2+^ (resp. 350 mg/L Ni^2+^) and 0.5 M citric acid. They indicate that at the chosen experimental leaching pH, Zn is in the form Zn(Citrate)_2_^4−^ and Ni in the form Ni(Citrate)_2_^4−^.

#### 2.4.1. Kinetics

Figure 11a shows the results of desorption of Zn in a 0.5 M citric acid solution, in the presence of ultrasound at 20/362 kHz or in silent conditions. The pH of the suspension was controlled and maintained between 5.5 and 6.5 throughout the experiment. Without ultrasound, Zn desorption is very rapid and a plateau is reached after 30 min. A desorption yield of 79% is attained under silent conditions. It is increased under US irradiation, especially at 362 kHz (88%). Comparing with Figure 4, it can be noticed that a better desorption is reached with citric acid than with hydrochloric acid in all conditions, thanks to the high ability of citric acid to complex Zn [32].

Corresponding curves with Ni are plotted in Figure 11b. As previously observed with HCl, the kinetics are much slower than for Zn and desorption yields are lower, reaching 10–13% after 3 h. In the Ni case, lower desorption yields are reached with citric acid than with HCl (17–28%), in spite of its high complexing ability. A slightly higher desorption yield is reached with 362 kHz US irradiation.

#### 2.4.2. Tessier Sequential Extractions

In order to gain a better understanding of the desorption mechanism, some clay samples were analysed by sequential extraction. The distributions of Zn and Ni in the different fractions are shown in Figure 12. After treatment with 0.5 M citric acid at 362 kHz, fractions F1 and F2 virtually disappeared for both metals. They also strongly decreased in other experimental conditions for Ni, showing the efficient complexation. In the case of Zn, fractions F3 and F4 also disappeared, and only a fraction (around 30%) of the metals present in the residual fraction in the initial state remained in this fraction after treatment. These decreases are in good agreement with results obtained on a sludge sonicated in citric acid solution [49].

In the Ni case, the F3 fraction also decreases, from 28% to 10–18%, but the F4 fraction increases a bit in most cases. As for the residual fraction F5, it is approximately constant. A negative impact of conditions favouring fragmentation (20 kHz and H-silent) is observed, with the increase in the F1 fraction. Reported results on a sludge submitted to washing in citric acid under US [49] are similar, with an approximate reduction by a factor of 2 of the F1–F3 fractions, though F4 was observed to decrease.

Comparing the results obtained with citric acid to those with HCl, the desorption yields are approximately half, indicating that for Ni, complexation is less efficient than acidity.

## 3. Materials and Methods

### 3.1. Materials

Vermiculite (Sigma Aldrich, Saint Louis, MO, USA, batch MKBJ9406V) was ground and sieved to obtain particles < 100 μm. Elemental analysis by acidic dissolution of raw vermiculite gave the following structural formula: (Si_2.63_Al_0.83_Fe_0.05_)(Fe_0.23_Mg_1.99_)O_10_(OH)_2_K_0.38_Ca_0.40_Mg_0.15_Na_0.19_.

The metal salts used to contaminate vermiculite and the salts/oxidants/acids used for leaching experiments and the Tessier sequential extraction protocol are listed in Table 1.

### 3.2. Sample Analysis and Characterization

X-Ray diffraction (XRD) patterns were recorded using a Malvern (U.K.) Panalytical X’Pert MPD Pro device with a Cu source (λKα1 = 1.5406 Å) in Bragg–Brentano geometry with a step of 0.013° and a counting time of 0.34 s. The deviation angle ranged between 5 and 70° and the measurement duration was 42 min. Particle size distributions of vermiculite samples before/after treatment were measured by laser granulometry with a CILAS (Orleans, France) 1090 Particle Size Analyzer. Inductively Coupled Plasma with Atomic Emission Spectrometry (ICP-AES) analyses (Thermo Scientific iCAP 6000 spectrometer, Waltham, MA, USA) were used to determine the concentration of metal pollutants (Ni and Zn) of liquid samples during desorption experiments.

### 3.3. Adsorption Isotherms and Kinetics

Kinetics adsorption experiments were carried out to estimate the duration after which thermodynamic equilibrium was reached. An amount of 20 mg of vermiculite was contacted with 20 mL of a 10 mg·L^−1^ Zn^2+^ or Ni^2+^ (from zinc nitrate or nickel nitrate) solution, containing also 0.01 M of NaNO_3_, for different durations (ranging from 10 min to 7 days) at ambient temperature. After shaking, centrifugation and supernatant filtration (through a Puradisc FP30 syringe filter, 0.2 μm), ICP-AES was used to measure the residual concentration in Zn^2+^ (resp. Ni^2+^) in solution. Vermiculite capacity Q (mg·g^−1^) was then determined for each contact time using Equation (1):(1)$$Q=\left(C_{i}-C_{f}\right)\frac{V}{m}$$
where C_i_ and C_f_ (mg·L^−1^) are the initial and final metal concentrations in solution, V (L) the volume of solution and m (g) the mass of vermiculite.

To calculate adsorption isotherms, 20 mg of vermiculite were contacted with 20 mL of solution containing 0.01 M of NaNO_3_ and a metal concentration between 1 and 1000 mg·L^−1^ for 7 days_._ Vermiculite capacity was then calculated as a function of metal concentration.

### 3.4. Preparation of Metal-Loaded Vermiculite

Several batches of zinc- and nickel-loaded vermiculite were prepared for ultrasound-assisted desorption from vermiculite ground and sieved < 100 µm (Appendix A). The aim of the preparation step was to obtain polluted vermiculite samples with a content of approx. 40 mg·g^−1^. It was reached using the isotherm adsorption protocol described in Section 3.3 and results obtained are given in Section 2.1. The pH was set at 4 in order to ensure that metal species were in ionic form. Batches B3.1(Zn), B3.2(Zn) and B3.3(Zn) were prepared by contacting 20 g of vermiculite for two weeks at ambient temperature to 2 L of solution with an initial metal concentration of 450 mg·L^−1^. For batches 3.4G(Zn) and 3.5G(Zn), 100 g of vermiculite were contacted for two weeks at ambient temperature to 1 L of solution with an initial metal concentration of 3200 mg·L^−1^. For batch B4.0(Zn), 100 g of vermiculite was contacted for one week at ambient temperature to 1 L of solution with an initial metal concentration of 450 mg·L^−1^. The amount of sorbed Zn^2+^ thus obtained ranged between 37.5 and 41.8 mg/g. As for Ni-polluted samples, batches B3.1(Ni), B3.2(Ni) and B3.3(Ni) were prepared by contacting 20 g of vermiculite for two weeks at ambient temperature to 2 L of solution with an initial metal concentration of 450 mg·L^−1^ and batches B3.4G(Ni) and B3.5G(Ni) by contacting 100 g of vermiculite for two weeks at ambient temperature to 1 L of solution with an initial metal concentration of 2300 mg·L^−1^. The amount of sorbed Ni^2+^ ranged between 30.2 and 37.4 mg/g. This variation in the sorbed amount is attributed to some heterogeneity in vermiculite samples.

Speciation diagrams of Zn(II) and Ni(II) in the presence of citrate/citric acid were calculated with the computer program PhreeqC interactive 3.7.3 (developed by the United States Geological Survey (UCGS)), using the Minteq V4 database.

### 3.5. Washing Experiments

Washing of contaminated vermiculite samples was performed using 0.1 M HCl alone or with MgCl_2_ 1 M, or 0.5 M citric acid solutions. The pH of citric acid solutions was fixed and kept constant at 5–7 during washing to guarantee innocuousness toward soil and biomass.

Washing was performed in two conditions of ultrasonic irradiation. At 20 kHz, a titanium sonotrode (Ti_6_Al_4_V, 1 cm^2^ cross section), connected to a 750 W generator (Sonics & Materials, Newtown, Connecticut, USA, Vibra-Cell VCX 750), was placed directly above the reactor with its tip immersed by 1 cm in the 200-mL liquid. At 362 kHz, the transducer (L3C ELAC Nautik, Kiel, Germany, 25 cm^2^), connected to a 125 W generator (T&C Power Conversion, Inc., Rochester, NY, USA), was placed at the bottom of a glass cylinder with coupling water, in which a second (smaller) reactor was inserted, containing 120 mL of sample. The distance between the bottom of the small reactor and the transducer was chosen to maximise transmission of the US wave. Unless otherwise stated, the ratio of vermiculite mass to washing solution volume was 20 g/L. To maintain its homogeneity, the suspension was continuously stirred at 570–760 rpm with a glass agitation blade placed vertically. At both frequencies, the bulk temperature was controlled using a Huber Ministat (Offenburg, Germany) 240-NR thermo-cryostat and kept around 20 °C. The bulk sample was continuously sparged with either Ar or air (it was checked that similar results were obtained in both cases) to keep the dissolved gas amount constant. The absorbed acoustic power, measured by the thermal probe method, was 34 W at 20 kHz and 30 W at 362 kHz.

To bring the impact of US to light, the same experiments were performed in silent conditions: in the 120-mL reactor with mechanical stirring with the glass agitation blade only (experiments called V-silent, Appendix A), and for comparison with set-ups commonly found in the literature, in a Heidolph (Schwabach, Germany) Reax 20 overhead shaker at 100 rpm (experiments called H-silent, Appendix A).

Aliquots of the suspension were sampled during the sonolysis with a syringe equipped with a filter, to monitor the concentration of metal in solution. At the end of each treatment, under US or in silent conditions, the remaining suspension was vacuum filtered before analysis of the solid and liquid phases.

### 3.6. Tessier Sequential Extraction Procedure

The evolution of how strongly metals were bound to the vermiculite was followed using the Tessier sequential extraction procedure [17] on 1 g of dry sample. Unless otherwise stated, the measured samples were either vermiculite samples prior to washing or vermiculite samples after a 3 h washing. Table 2 presents the experimental conditions of each step. After each step, the solid residue was rinsed with water and centrifuged at 9000 rpm during 10 min. The metal content of the supernatant was quantified by ICP and added to the corresponding step. The solid residue was submitted to the next extraction step.

Uncertainties (linked to ICP-AES accuracy) are of 14% for the metal removed in the decontamination experiments, of 17% for the metal concentration in F1–F4 and of 32% in F5.

## 4. Conclusions

This study investigated the contribution of ultrasonic irradiation in the desorption of two heavy metal pollutants, Ni and Zn, through a combination of acid washing and ion exchange. In the Zn case, for an m/V ratio of 20 g·L^−1^, very high desorption yields, around 85% after 3 h, were reached with HCl washing in silent conditions and under US: the presence of ultrasound at 20 or 362 kHz did not induce any significant improvement. However, the Tessier sequential extraction protocol showed that sonication can trigger the transfer of metal pollutants from the residual fraction to more accessible ones. Released cations readsorb in the exchangeable fraction, and a combined treatment with cation exchange leads to higher desorption yields. Sonication at low frequency and increasing the m/V ratio to 50 g/L leads to fragmentation of the vermiculite particles, and consequently to the creation of new surfaces presenting sorption sites for exchangeable cations, thus to an increase in the F1 fraction. Citric acid washing allowed to reach the highest desorption yields, in particular under US, which is promising due to its biobased nature.

Nickel extraction is more difficult due to its chemical speciation and adsorption in vermiculite, with a maximal desorption yield of 38%, obtained after 3 h of HCl-MgCl_2_ washing at 20 kHz. The use of ultrasound at 20 kHz allowed to improve access to Ni adsorption sites by internal diffusion and created new adsorption sites by particle fragmentation. Such phenomena induced an increase in the exchangeable fraction, which allowed improvement of nickel desorption by cation exchange with Mg^2+^. The impact of high-frequency US (362 kHz) is very different; it provoked a reduction in the F2, F3 and F4 fractions without noticeable changes in the exchangeable and residual fractions. Hence, a treatment combining successive 20 and 362 kHz cycles may be interesting to consider, due to the complementary effects.

This study brought to light for the first time the potential of high-frequency US in enhancing soil washing, to a higher extent compared to low-frequency (20–50 kHz) US. The methodology considered in this article remains to be applied on real soils to assess the potential of combination of acid washing and ultrasound.

## Figures and Tables

**Figure 1 molecules-30-01110-f001:**
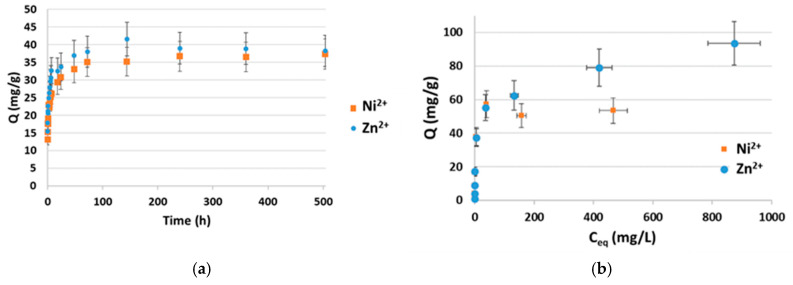
(**a**) Zn and Ni kinetics of sorption at ambient temperature—[metal] = 100 mg·L^−1^ with [NaNO_3_] = 0.01 M; m/V = 1 g·L^−1^. (**b**) Zn and Ni sorption isotherms at ambient temperature—[NaNO_3_] = 0.01 M; m/V = 1 g·L^−1^. C_eq_ denotes the concentration reached at equilibrium.

**Figure 2 molecules-30-01110-f002:**
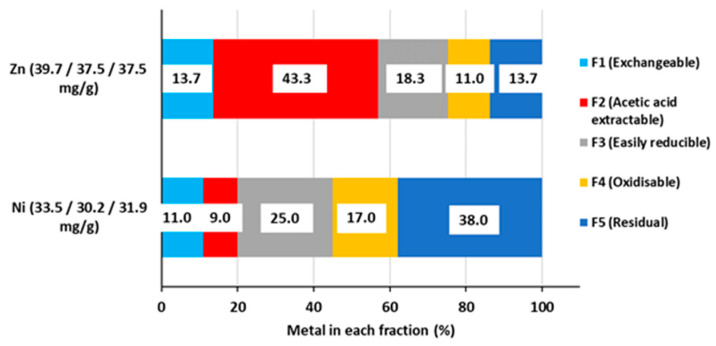
Zn/Ni distribution in vermiculite fractions obtained using the Tessier sequential extraction protocol (for each metal, average of three different batches, the metal load of each being indicated on the y-axis).

**Figure 3 molecules-30-01110-f003:**
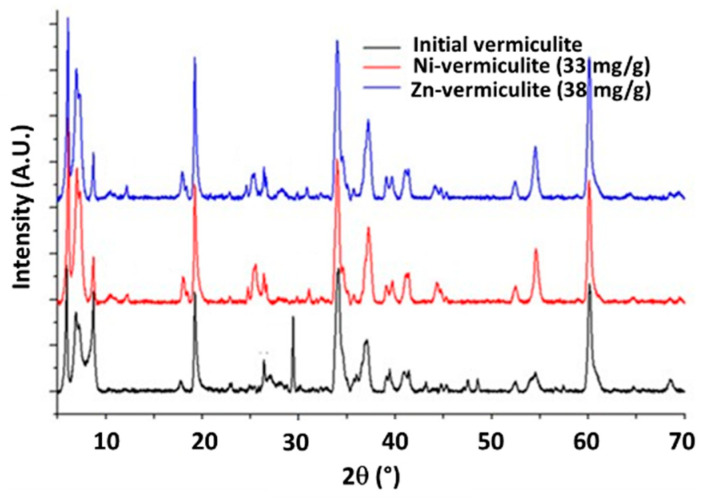
X-ray diffractograms of vermiculite samples, of initial composition and after loading with Zn^2+^ (38 mg·g^−1^) or Ni^2+^ (33 mg·g^−1^).

**Figure 4 molecules-30-01110-f004:**
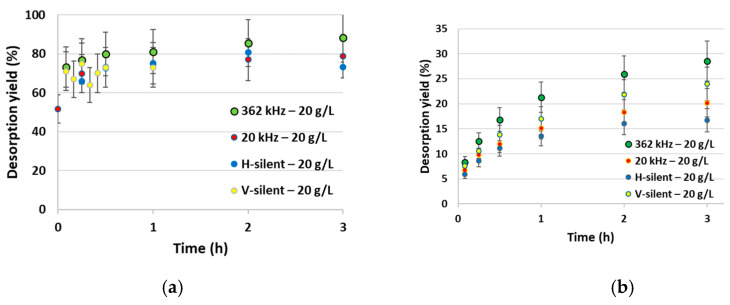
Zn (**a**) and Ni (**b**) desorption kinetics from contaminated vermiculite—influence of ultrasound frequency (silent conditions, 20 and 362 kHz); m/V = 20 g/L; HCl concentration = 0.1 M.

**Figure 5 molecules-30-01110-f005:**
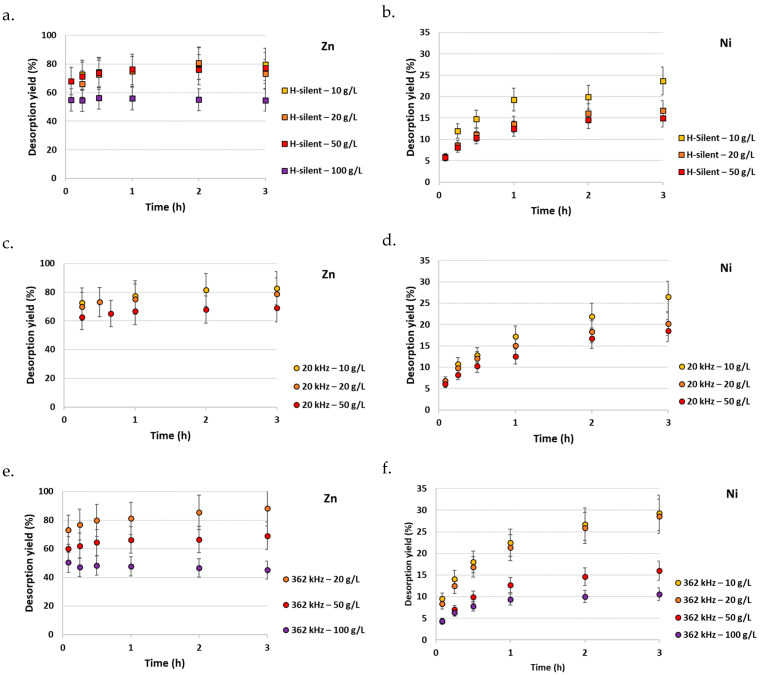
Influence of vermiculite mass to HCl solution volume (m/V) ratio on Zn and Ni desorption from polluted vermiculite; HCl concentration = 0.1 M. (**a**) Zn desorption under H-silent conditions; (**b**) Ni desorption under H-silent conditions; (**c**) Zn desorption at 20 kHz; (**d**) Ni desorption at 20 kHz; (**e**) Zn desorption at 362 kHz; (**f**) Ni desorption at 362 kHz.

**Figure 6 molecules-30-01110-f006:**
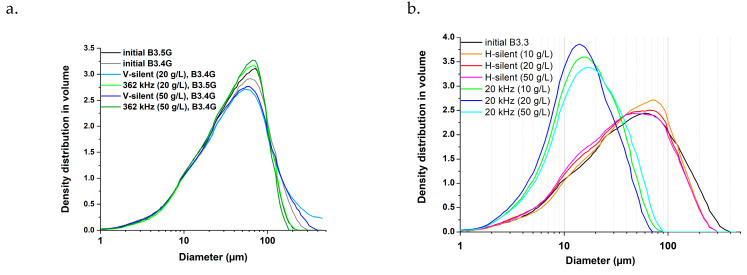
Particle size distribution of Zn/Ni polluted vermiculite before/after (**a**) V-silent or 362 kHz treatment; (**b**) H-silent or 20 kHz treatment. Properties of the used batches (B3.3, B3.4G and B3.5G) are presented in Appendix A.

**Figure 7 molecules-30-01110-f007:**
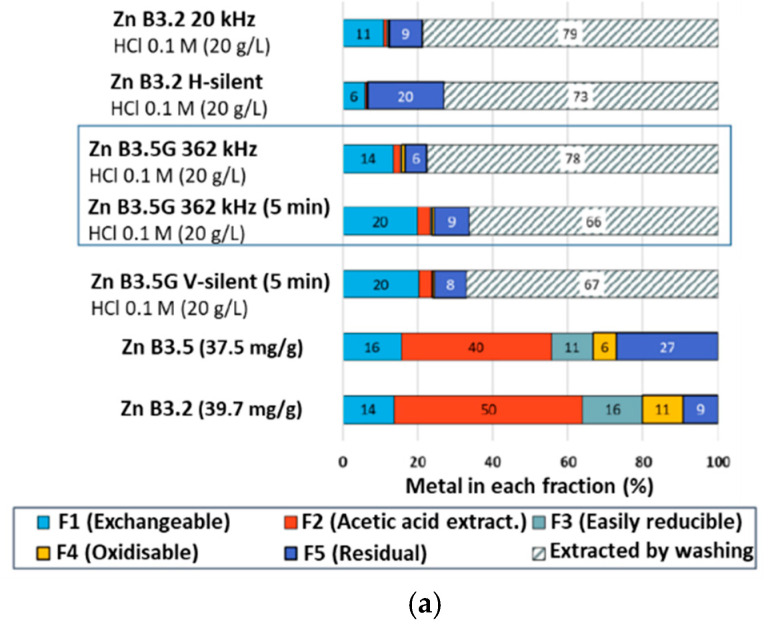
Evolution of Zn/Ni distribution in vermiculite fractions obtained using the Tessier sequential extraction protocol after washing with HCl 1 M during 3 h, under sonication or silent conditions: (**a**) Zn with m/V = 20 g/L; (**b**) Zn with m/V = 50 g/L, (**c**) Ni with m/V = 20 and 50 g/L.

**Figure 8 molecules-30-01110-f008:**
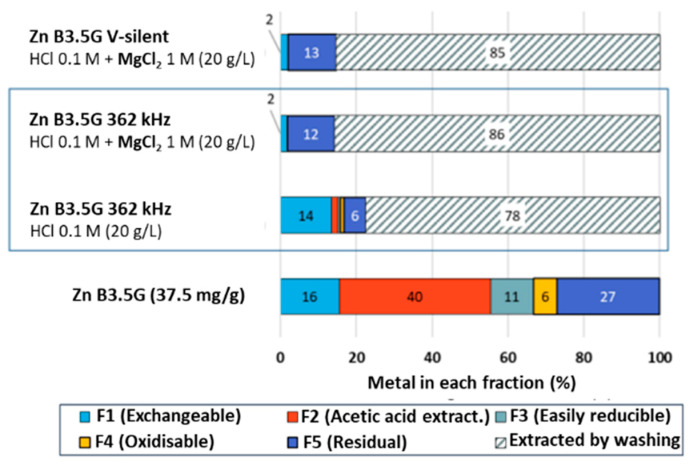
Evolution of Zn/Ni distribution in vermiculite fractions obtained using the Tessier sequential extraction protocol after washing with HCl (0.1 M) and MgCl_2_ (1 M). Desorption in V-silent conditions and at 362 kHz with m/V = 20 g·L^−1^.

**Figure 9 molecules-30-01110-f009:**
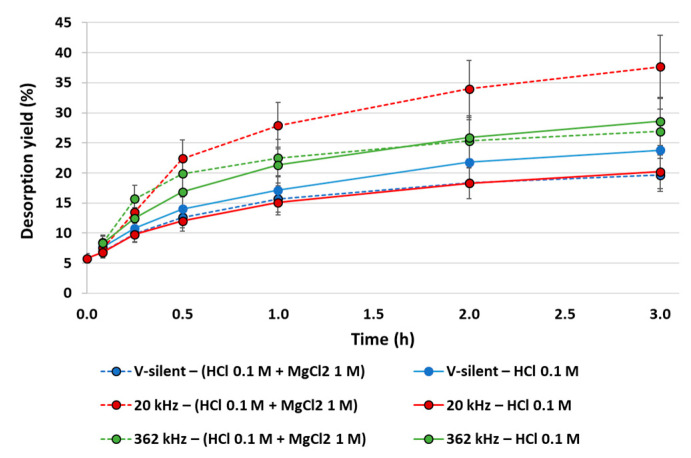
Ni desorption kinetics from polluted vermiculite in silent conditions, at 362 kHz and at 20 kHz–m/V = 20 g·L^−1^, HCl concentration to 0.1 M and MgCl_2_ concentration to 0 or 1 M.

**Figure 10 molecules-30-01110-f010:**
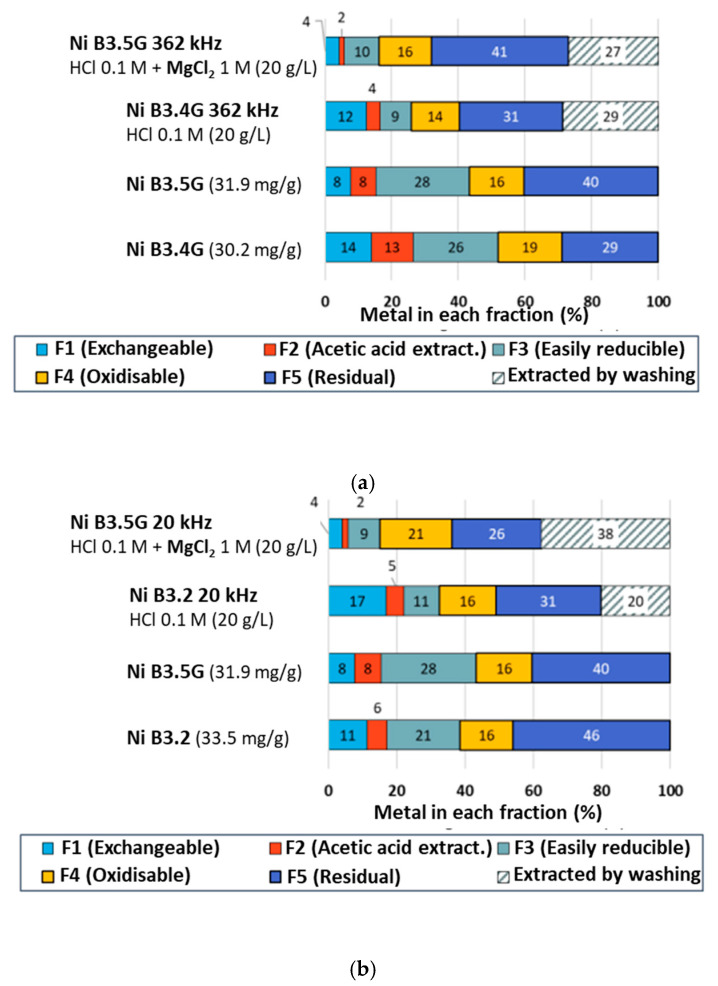
Comparison of Ni distribution in vermiculite fractions obtained using the Tessier sequential extraction protocol after washing with HCl 1 M + MgCl_2_ 1 M or HCl 1 M alone during 3 h: (**a**) under 362 kHz US irradiation; (**b**) under 20 kHz US irradiation. m/V = 20 g/L.

**Figure 11 molecules-30-01110-f011:**
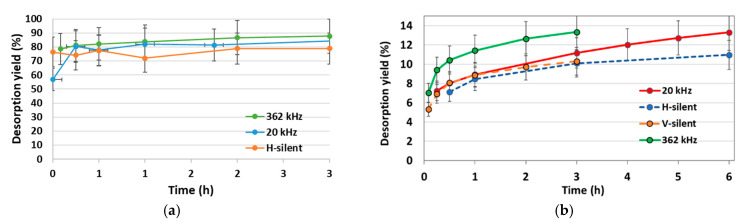
Zn/Ni desorption kinetics from polluted vermiculite in 0.5 M citric acid, under silent conditions or US irradiation; m/V = 10 g·L^−1^: (**a**) Zn; (**b**) Ni.

**Figure 12 molecules-30-01110-f012:**
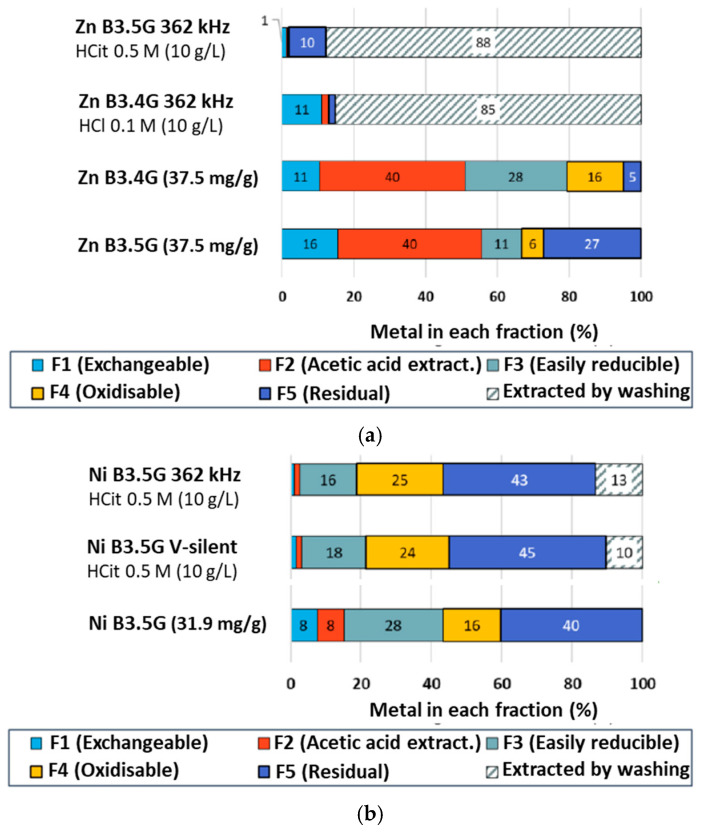
Comparison of Zn/Ni distribution in vermiculite fractions obtained using the Tessier sequential extraction protocol after washing with 0.5 M citric acid during 3 h: (**a**) Zn; (**b**) Ni. m/V = 10 g/L.

**Table 1 molecules-30-01110-t001:** Chemical structure and purity of the chemical products used in this study.

*Chemical Name*	*CAS N°*	*Source*	*Chemical Formula/Structure*	*Concentration or Purity (wt%)*
Acetic acid	64-19-7	Aldrich	CH_3_COOH	99.7
Hydrochloric acid	7647-01-0	Carlo Erba (Milan, Italy)	HCl	37
Citric acid	77-92-9	Aldrich	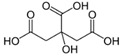	99.5
Nitric acid	7697-37-2	Aldrich	HNO_3_	70
Ammonium acetate	631-61-8	Thermo Fisher Scientific	CH_3_COONH_4_	99.99
Magnesium chloride	7786-30-3	Aldrich	MgCl_2_	99.99
Nickel (II) nitrate hexahydrate	13478-00-7	Aldrich	Ni(NO_3_)_2_·6H_2_O	99.999
Sodium acetate	127-09-3	Merck	CH_3_COONa	99.99
Sodium nitrate	7631-99-4	Aldrich	NaNO_3_	99.995
Zinc (II) nitrate hexahydrate	7779-88-6	Aldrich	Zn(NO_3_)_2_·6H_2_O	99.999
Hydrogen peroxide	7722-84-1	Aldrich	H_2_O_2_	30
Hydroxylamine	7803-49-8	Aldrich	NH_2_OH	50

**Table 2 molecules-30-01110-t002:** Experimental conditions of the steps of the Tessier sequential extraction procedure and corresponding extracted fractions.

	Reactants and Conditions	Fraction
F1	8 mL MgCl_2_ 1 M, pH 7, 2 h, 25 °C	Exchangeable cations
F2	8 mL CH_3_COONa 1 M, pH fixed to 5.0 with CH_3_COOH, 5 h, 25 °C	Acetic acid extractable fraction
F3	20 mL NH_2_OH.HCl 0.04 M in CH_3_COOH 25% *v*/*v*, pH fixed to 2 with HNO_3_, 6 h, 96 °C (in an autoclave)	Easily reducible
F4	(i) 5 mL H_2_O_2_ 30%, pH 2 + 3 mL HNO_3_ 0.02 M, 2 h, 85 °C (in a autoclave) (ii) then add 3 mL H_2_O_2_ 30%, 3 h, 85 °C (in a autoclave) (iii) after cooling, add 5 mL CH_3_COONH_4_ 3.2 M in HNO_3_ 20 vol% and complete to 10 mL with H_2_O, 30 min, 20 °C	Oxidisable
F5	Not measured but calculated by mass balance	Residual

## Data Availability

Data available on request.

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
