# Peer review of "Benefit of an Ultrasonic Irradiation on the Depollution by Washing of Nickel- or Zinc-Contaminated Vermiculite"

_molecules, 2025, doi:10.3390/molecules30051110_

Round 1
Reviewer 1 Report
Comments and Suggestions for Authors
The paper presents the effects of both ultrasonic irradiation and acid washing for the removal of heavy metals like Ni and Zn from contaminated vermiculite. The study presents interesting innovative approaches to soil remediation, however, the manuscript must be revised to addressed the following comments/suggestions/questions and to improve its readability:
- All abbreviations must be defined when first used in the text. Examples are US in the abstract, ICP-AES in section 2.2, Ceq in Figure 1, etc.
- Aside from the elemental analysis and partivle size distributions of vermiculite, what other characterizations were performed on vermiculite? Did the authors measured the surface charge? Does surface charge affect the readsorption of Ni at low frequency?
- The results are difficult to follow due to the use of a lot of different conditions with different codes/names used in the whole manuscript. A summary of all the experimental conditions/batches would be helpful to follow all the results.
- The acid washing involved HCl and citric acid solutions, why is the pH fixed for the citric acid solutions but not for HCl?
- Paragraph 1.2 have been mentioned several times describing the adsorption protocol but reviewing this paragraph doesn't mention any protocol. What exactly are the authors refering to with Paragraph 1.2?
- Figure 2, what is the values presented in the y-axis? The values presented are a bit confusing.
- Decimal points should be presented with a dot and not with a comma.
- For most of the desorption yields and metal fractions, do the authors performed statistical tests? Most of the error bars are overlapping. Also, x-axis (mostly time) have error bars too, are the sampling times performed arbitrarily within the specified time periods?
Reviewer 2 Report
Comments and Suggestions for Authors
The study examined a method for the removal of heavy metals, concentrating on the desorption of nickel and zinc from vermiculite by a combination of leaching and ultrasonic irradiation at frequencies of 20 or 362 kHz. The authors delivered a thorough analysis and provided clear explanations. The subsequent inquiries and amendments should be handled.
Q1- What prompted the authors to choose a frequency of 362 kHz, and what criteria influenced their selection?
Q2- An illustration of the experimental setup should be provided to elucidate the installation of the ultrasonic apparatus for better understanding of the experimental setup.
Q3- Have writers contemplated the observation of ultrasonic vibration effects at a designated ultrasonic frequency?
Q4- Figure 3 should be edited for better visualization.
Q5- Citric acid washing allowed to reach the highest desorption yields, 526 in particular under US, which is promising due to its biobased nature. [Could you explain further the high deposition yields connected with the biobased nature ?]
Reviewer 3 Report
Comments and Suggestions for Authors
Dear authors!
I have read the Manuscript and I recognise that a very big work was carried out. As there is a lot of experimental data, it is always a challenge to present it clearly and make it understandable for the reader, as well as highlight the important parts. I have attached the file with some comments to the text, including some typos that I was able to notice, etc.
My major concern in this Manuscript is that some of the terminology is not explained properly (e.g., I was not able to find descriptions of what V-silent and H-silent modes are). Besides, there are some inconsistencies between the sections. For example, different standart concentrations of vermiculite suspencion were used for the experiments using HCl (20 g/L) and citric acid (10 g/L). At the same time it was shown that concentration of the suspension affects the extraction process. So, it is not quite correct to compare these two sections, or it must be done with caution and with extra explanations and precise references to Fig. from section 3.2
Besides, the Manuscript is poorly formatted, and does not follow the MDPI style guide. A lot of the images need to be re-made, as well as the referenced.
I hope that these changes will be made and look forward to reading the revised version of the Manuscript.

The Engllish language is fine and does not impede the understanding of the text. However, sometimes the phrasing is a bit confusing and some minor languge editing will be benificial.
Round 2
Reviewer 1 Report
Comments and Suggestions for Authors
The revised manuscript sufficiently addressed all my comments, hence, I recommend to accept the paper for publication in Molecules.
Reviewer 3 Report
Comments and Suggestions for Authors
Dear authors!
Thank you for all the improvements made to the Manuscript. I see that all the Reviewers' comments were answered in detail. Unfortunately, the image resolution remains very low, which is not suitable for publication. It might be a technical issue: lately I had cases when separately uploaded high-res images were not integrated into the text correctly be the submission system. Please figure out the reason for poor image quality and fix it. Otherwise I believe the work can be accepted for publication.